# New Technique for Monitoring High Nature Value Farmland (HNVF) in Basilicata

Costanza Fiorentino [1,*], Paola D'Antonio [1], Francesco Toscano [1], Angelo Donvito [2] and Felice Modugno [1]

1    School of Agricultural, Forestry, Environmental and Food Sciences, University of Basilicata, 85100 Potenza, Italy
2    DIGIMAT S.P.A., 75100 Matera, Italy
*    Correspondence: costanza.fiorentino@unibas.it

**Abstract:** The definition of High Nature Value Farmland Areas (HNVF) was provided by Andersen in 2003: "HNVF comprises those areas in Europe where agriculture is the major (usually the dominant) land use and where that agriculture supports or is associated with either a high species and habitat diversity, or the presence of species of European conservation concern or both". The present work focuses on an overview of the techniques used to produce HNVF maps at different spatio-temporal resolutions. The proposed approach is based on the statistical approach. The study area is the Basilicata region (southern Italy) in 2012, mapped at municipal spatial resolution. The HNVF areas were identified by applying a threshold to the sum of the contributions of the main characterizing indicators. Three indicators contribute to the calculation of the HNVF areas: crop variability (CD Index), extensive practices (EP Index), and the presence of natural elements (Index Ne). Good agreement was found between our HNVF map and the results of the literature, although the analysis approaches were different. The main advantages of the proposed methodology derive from only free input data being used, and include remote sensing images and the adaptability to different spatial resolutions (local, regional, and national).

**Keywords:** biodiversity; sustainability; MODIS; GIS; bioeconomic; big-data

## 1. Introduction

The elements that characterize agricultural areas that can be defined as having High Naturalistic Value [1–3] are areas characterized by a good level of crop diversity, where machinery and inputs such as fertilizers and pesticides are reduced; semi-natural areas with extensive agriculture; and the presence of hedges, rows of trees, and areas of spontaneous vegetation [4,5].

HNVF areas contribute significantly to the preservation and maintenance of a high degree of biodiversity. The work of Andersen et al. [6] identified potential HVNF areas on a European scale by combining the cartographic information of the Corine Land Cover (CLC) with the statistical economic information of the Farm Accountancy Data Network (FADN).

Since the first studies, three types of HNVF areas have been defined:

Type 1: Agricultural land with a high coverage of semi-natural vegetation.

Type 2: Agricultural land dominated by low-intensity agriculture or by a mosaic of semi-natural and cultivated territories.

Type 3: Agricultural land with rare species or a high proportion of animal and/or plant species of conservation interest.

The characterization of the three types of HNVF areas and how they can be operationally identified within agricultural areas is shown in Figure 1.

HNVF acquired particular importance in 2005 when it was adopted as an indicator by the Common Monitoring and Evaluation Framework of the Rural Development Programs [7,8]. Although the European Commission [9–12] did not provide a detailed and rigorous methodology for HNVF identification, it indicated general directives, leaving the

possibility of adapting the methodologies to the different geographical areas and to the available data [13].

| Methodology | HNVF Type 1 | HNVF Type 2 | HNVF Type 3 |
|---|---|---|---|
| Land Cover (Corine LC) | Presence of categories of CLC linked to HNVF<br><br>Indicative maps on the localization of HNVF | Presence of categories of CLC linked to HNVF<br><br>Indicative maps on the localization of HNVF | Not Applicable |
| Farm system survey<br><br>FDAN (Farm Accountancy Data Network)<br>RICA (Structural Data and Economic Indicators Network) | Presence and extensions of farms within the HNVF<br><br>Indicators of preccure on HNVF at farm level | Presence and extensions of farms within the HNVF<br><br>Indicators of preccure on HNVF at farm level | Not Applicable |
| Species and Habitat (Natura 2000, IBA, PBA, IPA) | Presence of secies and habitats<br><br>Geographic maps | Presence of secies and habitats<br><br>Geographic maps | Species and habitat distribution maps show relationships with different approaches and aid in the identification of other HNVF areas |

**Figure 1.** Scheme of the three types of HNVF areas and how they can be operationally identified within agricultural areas.

The European Evaluation Helpdesk for Rural Development has published a methodology to monitor HNVF areas at different spatial resolutions. This requires the analysis to be:

1. Developed and stored in a GIS project by using geo-referenced data and maps.
2. Based on integrated methods on land cover and cultivation/breeding intensity, taking into account the elements that preserve the distribution of species.
3. Dynamic—-in order to monitor the spatio-temporal variation of the HNVF areas.
4. Able to record and report an increase or a decrease of the HNVF indicators, and, therefore, the level of biodiversity of a specific area.

The 2020 Common Agricultural Policy (CAP) reform confirms the attention that has been given to environmental sustainability, biodiversity, and landscape, and underlines a fair income for farmers. The main changes of the post-2020 CAP concern the ways in which member states will determine how to achieve objectives and targets, including those for semi-natural and agricultural areas that are declining or at risk of declining, as well as their associated farmland species [14].

The new reform of the CAP for the period 2023–2027 will require countries to develop national strategic plans to promote the environmental and social sustainability of agricultural systems. This plan will indicate to the competent authorities the specific actions to be implemented, and it will specify the funds allocated and the evaluation parameters.

This conclusion came following careful assessments presented to the Agriculture Commissioner. Reports of workshops and scientific expert groups from European countries [15,16] emphasized the urgent need to:

- Increase protected areas;
- Increase funding to mitigate the negative effects of agriculture on biodiversity and climate;
- Increase and optimize funds that finance environmental and socio-economic objectives.

The new CAP aims to foster a sustainable and competitive agricultural sector that can support the livelihoods of farmers and provide healthy and sustainable food for society as well as rural areas. Agriculture and rural areas are central to the European Green Deal, and the new CAP will be a key tool in reaching the ambitions of the Farm to Fork and biodiversity strategies.

Actions should be planned both at territorial and farm level, and should be aimed at preserving and promoting biodiversity in relation to agricultural practices.

Actions should be based on the conservation of those agricultural areas that have natural elements supportive of biodiversity, such as agroforestry areas or landscape elements either bordering or within fields, such as isolated trees, stone walls, streams, or ponds that can host protected species.

Other important elements concern agronomic management starting, from the management of the field margin and therefore the adjustment of the dimensions of the field to arrive at the management of water resources, fertilizers [17], and the use of agro-chemical products for weed control and disease prevention. Managing all of these landscape and management elements of the agricultural environment is quite complex, and the possible solutions differ between agricultural systems and within agricultural systems depending on the geographical location [18,19].

With regards to conserving biodiversity, other elements play a role, such as managing fallow in arable land, maintaining grass cover in woody and mixed crops, flood control in paddy fields, and managing stocking rates in pasture, especially in the extensive ones. Tarjuelo et al. (2021) [20] showed that actions such as reducing pesticide use improved food availability and, therefore, bird diversity of field-scale farmland, while delaying the harvest offers more food and more shelter by improving bird abundance.

The techniques and methodologies for the recognition of HNVF support the conservation of biodiversity. Since their elaboration is based on the synthesis of multiple indicators, they can provide valuable ecosystem services to society, contributing to both sustainability and resilience in Europe [21]. HNVF calculation procedures can provide information about primary production, nutrient cycling, soil formation, etc. These indicators feed into a wide range of ecosystem services. HNVF agricultural land has a positive impact on climate change, soil erosion prevention, and biological control [22]. HNVF soils contain higher levels of organic carbon, underscoring their potential contribution to regulating climate, maintaining soil fertility, and preventing soil erosion, desertification [23,24], and salinization [25].

The HNVFs assume a supporting role for cultural services that is recognizable and economically valuable for citizens [21,26]. The supply of agricultural products of high quality and high economic value is linked to HNV agricultural land [26]. These products are often labeled as being of recognized high quality [27]. The preservation of HNVF areas, therefore, indirectly affects the quality of food production and livestock fodder, water supply, and cultural services, such as recreational activities, agro- and eco-tourism, the maintenance of cultural heritage, and scenic landscape [28].

The main objective of the present work is the definition of GIS procedures based on the elaboration of specific indicators in order to elaborate the HNVF map by using the Big Data available in the repository of information of the public authority database.

The proposed methodology has the advantage of using free input data (e.g., the regional orthophotos, the national and regional statistical information, and the maps of Corine Land Cover (CLC), Remote Sensing images, etc.), and it is adaptable to different spatial resolutions (local, regional, and national).

## 2. Materials and Methods

### 2.1. The Study Area

As shown in Figure 2, the study area is the entire Basilicata region at municipal level spatial resolution. The study year was 2012.

The Basilicata region is characterized by a mainly agricultural economy. It ranges between very different ecosystems: the south-west part is rich in woods and natural areas; the coastal strip has vineyards, orchards, and horticultural crops; and the inland areas have a prevalence of wheat cultivation. In the north-eastern area of the region, there are many industrial installations.

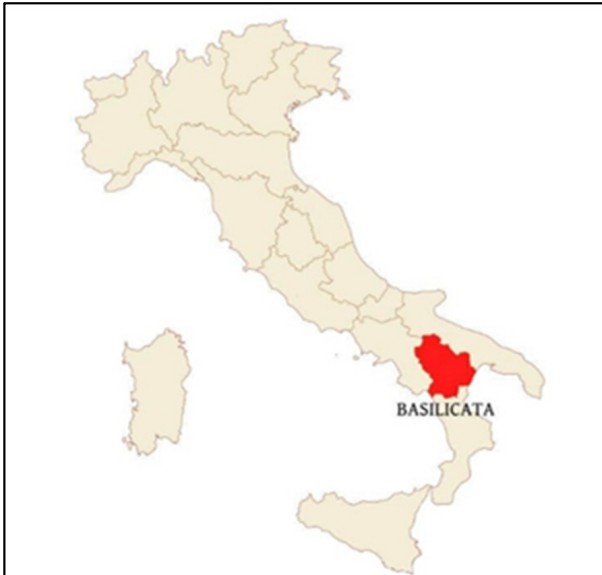

**Figure 2.** Geographic location of Basilicata region in southern Italy.

The input data used to elaborate the map of the HNVF areas are listed below, and the data were divided into three groups and uploaded in a GIS project (QGIS) [29]:

1.  Landscape conformation and structure:

    *   DEM/DTM computed from remote sensing SAR images (http://rsdi.regione.basilicata.it/; accessed on 5 April 2018).
    *   Topographic maps (http://rsdi.regione.basilicata.it/; accessed on 5 April 2018).
    *   Map of coastal areas and dunes covered by vegetation.

2.  Land Use

    *   Corine Land Cover map (CLC).
    *   Modis Satellite Images.
    *   Orthophoto 2012 (http://rsdi.regione.basilicata.it/; accessed on 5 April 2018).
    *   Map of protected areas: National and Regional Parks, SPAs, SIC, and Habitat map.
    *   Map of DOP, IGP, and organic crops.
    *   Vulnerability maps (http://rsdi.regione.basilicata.it/; accessed on 5 April 2018).
    *   Zoning map (2007–2013 RDPs) dividing the Basilicata territory into 3 homogeneous areas. This provides information on the degree of agricultural specialization, and indirectly provides information on the intensity of external inputs.

3.  Statistical data

    *   Data from the 6th agricultural census (ISTAT, http://www.istat.it/it/censimentoagricoltura/; accessed on 5 April 2018).
    *   FDAN Farm accountancy data network.
    *   RICA Structural data and economic indicators network (Italian CREA).

*2.2. Methodology*

The methodology is based on the statistical and farm systems approach. The originality of the developed procedure lies in the use of MODIS [30] satellite images:

-   To improve the number and accuracy of the land cover classes of the Corine Land Cover map.
-   To calculate indicators aimed at monitoring soil and vegetation properties.

Three independent indices contribute to the characterization of the HNVF area:

1.　Crop variability (CD Index).
2.　Extensive practices (EP Index).
3.　Presence of natural elements (Index Ne).

Each index summarizes the main characteristics of an area of high naturalistic value, and the criterion for defining whether a municipality can be classified as an HNVF area is as follows:

$$\text{Index HNVF} = (\text{CD} + \text{EP} + \text{Ne}) > \text{Threshold}; \tag{1}$$

According to the literature [31–35], the accepted minimum threshold to qualify an area as HNV farmland should oscillate between the 30th and the 15th percentile of the best municipality scores. The methodological framework was designed to identify municipalities whose utilized agriculture area (UAA) is mostly HNV. The detailed calculation of the indicators is shown in Table A1. The three indices (CD, EP, and Ne) were given equal weight in the calculation of the HNVF index, and they were normalized to vary between 0 and 10 in order to calculate a final score and to draw maps so that the total HNVF index varies between 0 and 30.

The indicators refer to the Utilized Agricultural Area (UAA). The UAA is calculated by excluding the man-made areas, the stretches of water (including rivers and canals), and the coastal areas of dunes covered by vegetation from the total municipal area of the woods. The study includes all 131 municipalities of the Basilicata region.

Remote Sensing MODIS Images

The Terra and Aqua combined Moderate Resolution Imaging Spectroradiometer (MODIS) images are available on the Earthdata website (https://earthdata.nasa.gov//; accessed on 5 April 2018).

From the website, it is possible to download a series of thematic maps (e.g., land cover, vegetation indices, land surface temperature, etc.) derived from MODIS multispectral images by using specific algorithms available in the user guide. In particular, two products were used in this work:

1.　Land Cover Type (MCD12Q1): The Version 6 data product provides global land cover types at yearly intervals (2001–2019) and at the spatial resolution of 500 m. Land cover types are: 11 classes of natural vegetation and 3 classes of barren or without vegetation, while the remaining 3 comprise a mixture of different types and/or artificial vegetation, such as croplands, The 2010, 2011, and 2012 land cover maps were used to refine and integrate the CLC map of 2012, while the 2012 orthophoto of the Basilicata region was used to validate the final product. An overall accuracy [36] of about 85% was reached.
2.　The Aqua/MODIS Land Surface temperature LST dataset (MYD11A2, version 006) starts from 2003. MYD11A2 actually contain the latest improvements to enhance estimation accuracy of LST. The spatial resolution of MYD11A2 is 1 km$^2$.
3.　MODIS vegetation indices: Normalized difference vegetation index (NDVI) [37] is derived from atmospherically-corrected reflectance in the red and near-infrared wavebands, and it is useful to more effectively characterize the global range of vegetation states and processes.

The daytime of LST and NDVI maps were collected at monthly intervals during the 2012 study year at the spatial resolution of 1 km. From LST and NDVI data, the Soil Moisture Index (SMI) [38] was computed:

$$\text{SMI} = \frac{(\text{LST}max - \text{LST})}{(\text{LST}max - \text{LST}min)}; \tag{2}$$

where LST*max* and LST*min* are the maximum and minimum surface temperature for a given NDVI, and where LST is the surface temperature of a pixel for a given NDVI derived using remote sensing data. LST*max* and LST*min* are calculated using the following equations:

$$\text{LST}max = a_1 * \text{NDVI} + b_1 \text{ and } \text{LST}min = a_2 * \text{NDVI} + b_2 \tag{3}$$

where $a_1$, $a_2$, $b_1$, and $b_2$ are the empirical parameters obtained by the linear regression (a present slope and b present intercept) defining both warm and cold edges of the data [38]. In Figure 3, the monthly SMI index maps from May to September 2012 are shown:

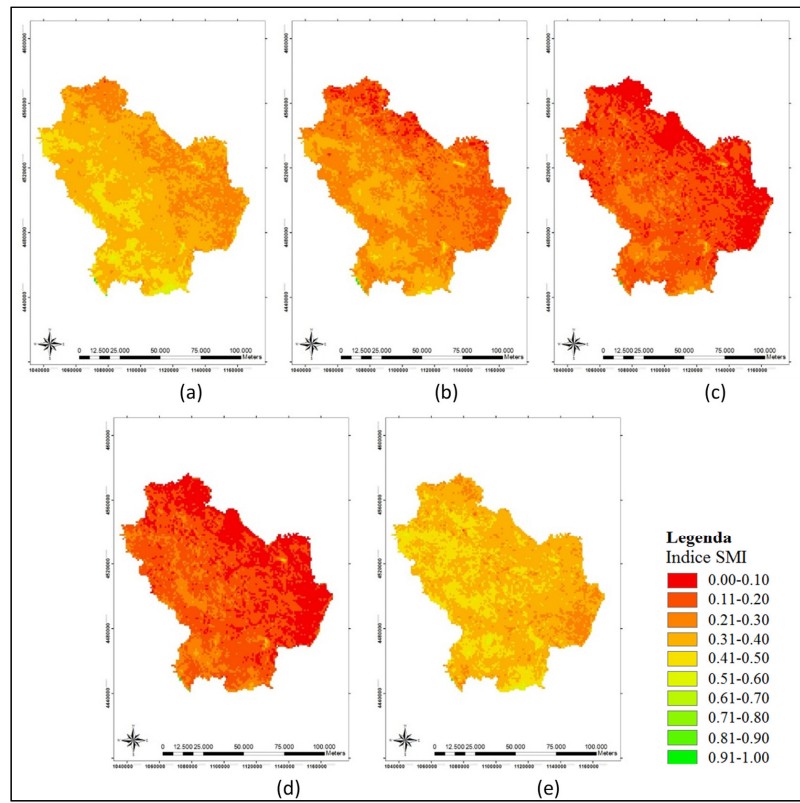

**Figure 3.** Monthly SMI index maps of May (**a**), June (**b**), July (**c**), August (**d**) and September (**e**) 2012.

## 3. Results and Discussion

In the literature, there are many examples of HNVF calculation in different areas of Europe with very different methodologies, both for the type of data and the processing techniques [39,40]. In some cases, biodiversity indicators relating to the presence of animal or plant species to be preserved are not involved in the analysis [41]. The type of indicators processed significantly affects the HNVF identification, and, moreover, the use of several methods can lead to widely diverging results.

In the Marche region (Italy), HNVF were recognized on the basis of vegetation structure and characteristics [42], and in France, Pointerau [34] used only statistical information, and their work was refined in 2010 by introducing additional vegetation information [35].

Morelli et al. (2014) studied the relationship between HNVF areas and breeding bird species, highlighting that HNVFs generally coincided with agricultural mosaics but did not include areas of conservationist species linked to less diverse agricultural landscapes.

Many studies suggest that identifying HNVF excluding extensive agricultural landscapes results in only heterogeneous landscapes being chosen in addition to natural habitats used for livestock (meadows and semi-natural pastures).

The biodiversity indicator is often calculated exclusively using data relating to the distribution of species [43], based on information from the Natura 2000 network. However,

this map neglects the species that live in agricultural environments, especially the extensive ones. In the Mediterranean environment, these habitats are predominant and therefore very important for the conservation of species [44].

The proposed approach is based on the statistical and farm system approach, and all indicators in Table A1 have been calculated in QGIS [29], starting from the input data.

The HNVF areas were identified according to the criterion: HNVF Index > Threshold: the threshold goes from 18.34, the minimum value corresponding to the 30th percentile of the HNVF Index of the municipality scores, and the maximum value corresponding to the 15th percentile equal to 19.27.

Figure 4 shows the three indices, CD, EP, and Ne, and the HNVF areas are characterized by greater crop variability and the greater presence of natural elements that favor biodiversity. In these areas, there is a lower coverage of extensive crops, but this is due to the greater crop variability. The index relating to extensive farming does not vary significantly between the two macro-areas and, therefore, does not seem to affect the final result. On the other hand, the average value relating to the presence of extensive meadows and pastures is much lower in the areas classified as non-HNVF. The soil moisture index is on average higher for HNVF areas, while the nitrogen supplied to crops but not used is lower in these areas.

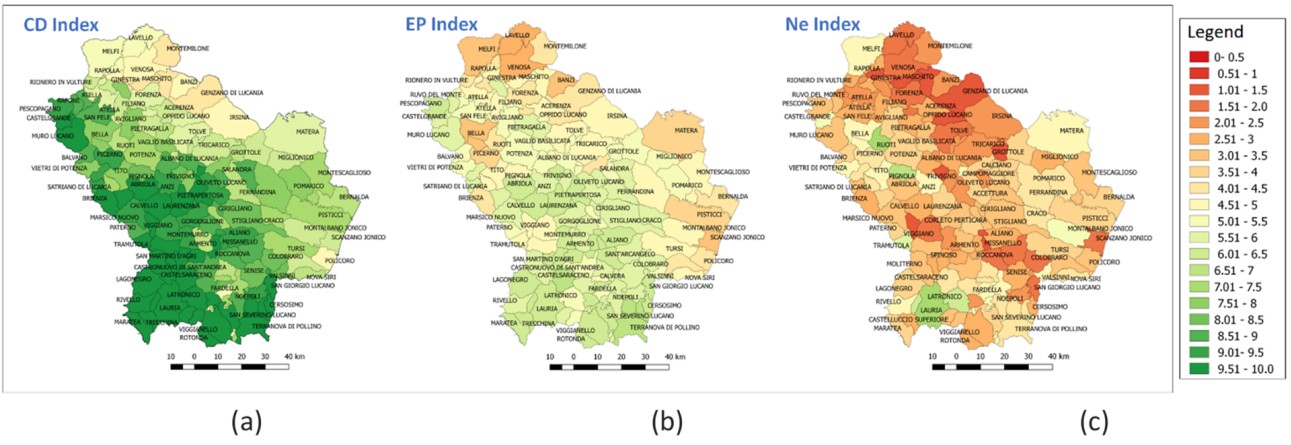

**Figure 4.** (**a**) Map of the Crop Diversity Index (CD); (**b**) Map of the Extensive Practices index (EP); (**c**) Map of the presence of Natural Element index (Ne).

In Figure 5, the map of HNVF index is shown.

Figure 6 shows the maps of classified HNVF areas (in green) obtained using a threshold value equal to the 30th percentile (Figure 6a) and the 15th percentile (Figure 6b). The threshold at the 30th percentile seemed the most appropriate, taking into account of the protected areas map of the Basilicata region.

As can be seen from Figure 6a, 39 municipalities were classified as agricultural areas with a high naturalistic value, and the remaining 92 were excluded.

Table A2 lists the 13 municipalities that fall between the 30th and 40th percentile of the HNVF index. The HNVF index associated with the municipality of Salandra (classified as non-HNVF) is the closest to the municipalities classified as HNVF. Salandra has a CD index value lower than the average value of the HNVF class, an EP index comparable to the average value of the HNVF class, and an Ne index slightly lower than the average value of the HNVF class. Concerning the EP index, the municipality of Salandra has a higher level of extensive crops than the average of the HNVF class as well as the index relating to extensive farms, but it lacks natural meadows. Therefore, the municipality of Salandra should increase crop variability and/or increase the area of extensive meadows and pastures in order to re-enter the HNVF area, but only 4% of the UAA is intended for non-extensive meadows and pastures. The uncultivated areas contribute to the EMC index associated with the presence of extensive crops, which in 2010 for the municipality

of Salandra represented 35% of the total UAA compared to 35% of the UAA intended for agricultural crops, of which extensive crops accounted for 15%. In conclusion, the municipality of Salandra should maintain and increase the agricultural area dedicated to extensive crops in order to be able to return to the HNVF area.

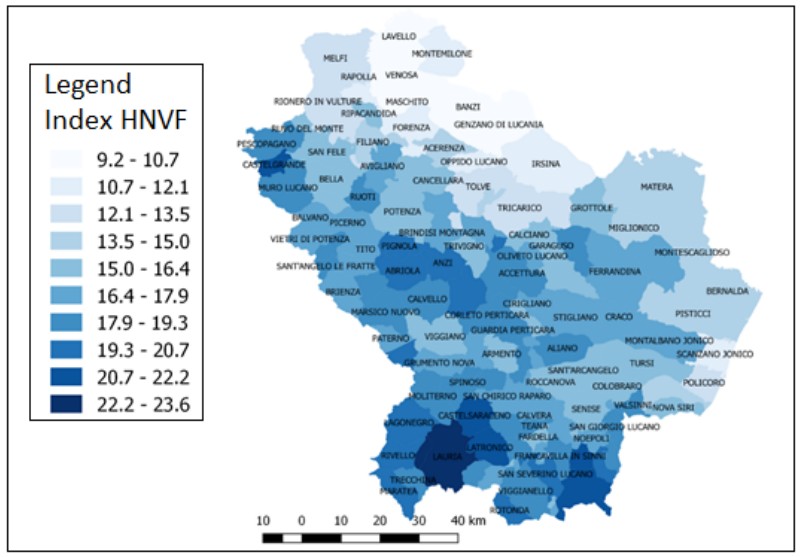

**Figure 5.** Map of HNVF index obtained as the weighted sum of the three sub-indexes CD, EP, and Ne. The index vary between 0 and 30.

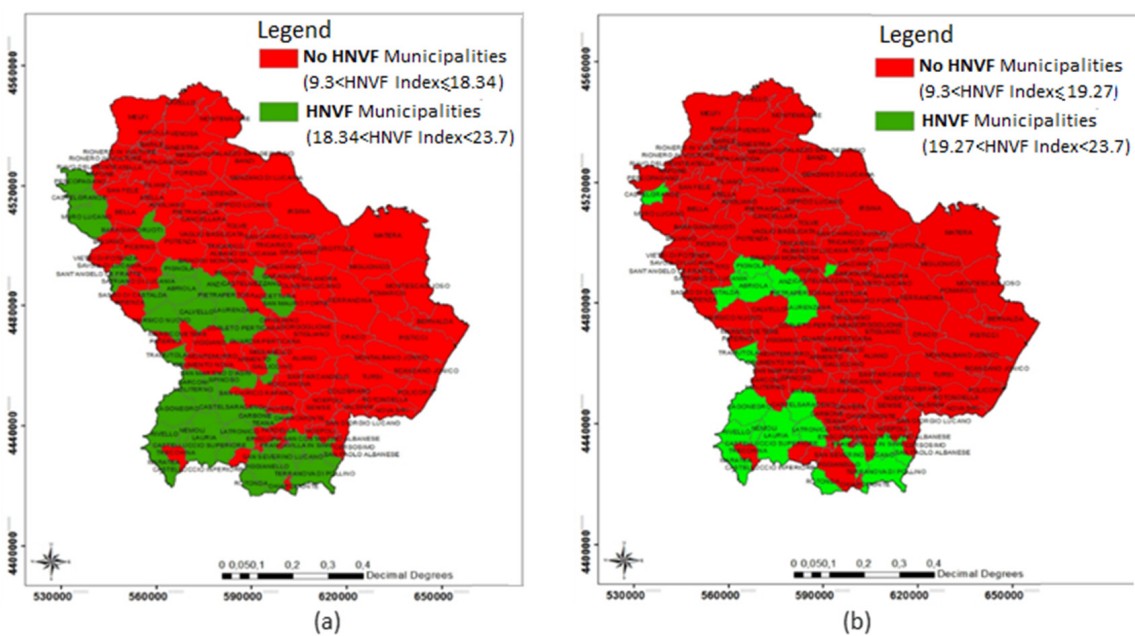

**Figure 6.** HNVF Map of the municipalities in Basilicata region. Figure (**a**) shows the map relating to a threshold value equal to the 30th percentile, while Figure (**b**) shows the map relating to a threshold value equal to the 15th percentile.

In 2017, ISPRA [45] published a detailed document in which the HNVF index (Figure 7) was calculated based on three elements: indicator of conservation areas of interest, geodiversity indicator, and anthropic impact indicator. The first was calculated starting from the presence on the territory of areas of naturalistic importance and conservation interest with respect to the total area of the landscape unit, and the second represented, for each landscape unit, the total number of geo-sites and natural monuments present in the area.

For the calculation, only sites with a degree of regional, national, or international interest were considered, and those of local interest were excluded. The last indicator contributes to the calculation of the natural value of each landscape unit by deducting the disturbance due to the presence of artificial environments from the maximum potential natural value. Two different detractors were considered: the population density per square kilometer and the constraint of naturalness (e.g., the disturbance due to the adjacency of highly man-made environments).

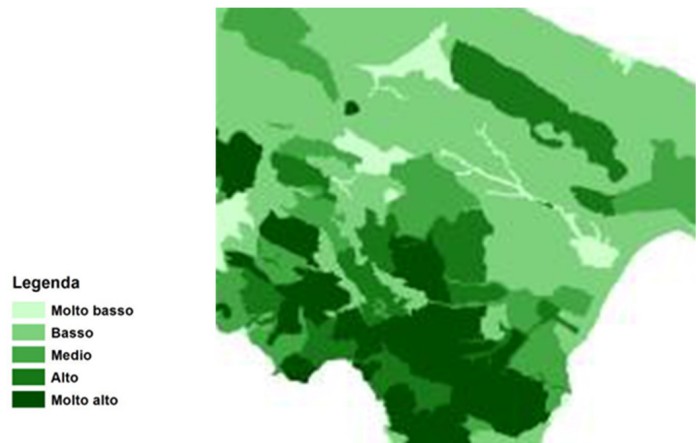

**Figure 7.** Map of Natural value index in Basilicata region (ISPRA 2017).

In 2015, Cozzi et al. [46–48] used, for the construction of the HNVF area identification model, a Multi Criterial Evaluation (MCE) procedure with a standardization of the criteria according to the fuzzy logic (Figure 8a). The integrated multi-criteria model for the identification spatial areas HNVF enabled the integration of seven criteria: land use, Rural Development plan areas, protected areas, vulnerability soils, hydrography, surface organics, and DOP and IGP products. The territorial indicators were calculated in a GIS, and the comparison between the various factors was made through a Hierarchy Analytics Process (AHP). Four different matrices of judgments were built, one for each category of decision makers involved, according to the opinions of farmers, environmental associations, policy makers, and technical operators. A large part of the agricultural land of Basilicata (about 48%) has a high conservation value.

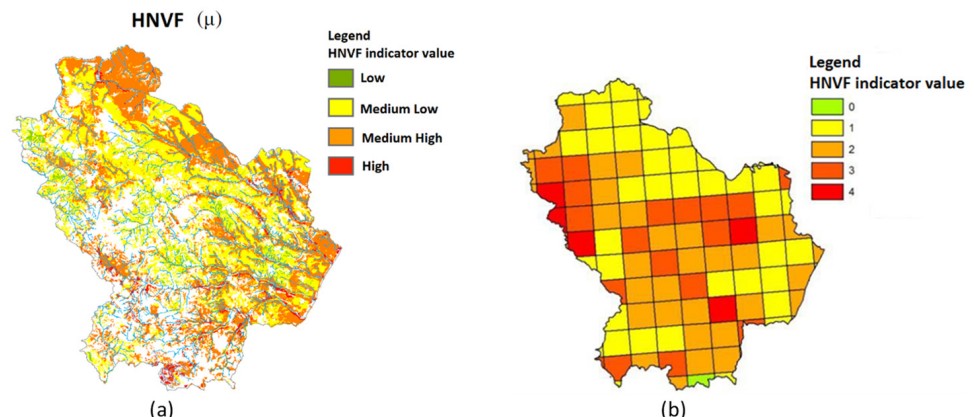

**Figure 8.** Maps of HNVF areas in Basilicata region published in 2014 by Cozzi et al. [46] (**a**) and in 2014 by De Natale et al. [4] (**b**).

In 2014, De Natale, Pignatti, and Trisorio mapped the HNVF areas in Basilicata (Figure 8b) at the spatial resolution of 100 km$^2$, using the land cover approach. In particular, three criteria were followed in order to identify high-value agricultural areas: a high

proportion of semi-natural vegetation; the presence of natural, semi-natural, and structural elements of the landscape; and the presence of species of interest for nature conservation at the European level.

Only a qualitative and non-quantitative validation was made due to the large difference between the processing methodologies used to identify HNVF areas. The map in Figures 5, 7 and 8b shows the presence of a similar pattern. In fact, the areas associated with highest values of the HNVF index are located in the center of Basilicata and in a strip in the southwestern region, while in the north-east of the Basilicata region, there are many industrial facilities. The map in Figure 8a appears quite different from the others, but it was calculated using the approach of the protected species that is very different from the other methodologies [49].

## 4. Conclusions

A primary objective of the new CAP is the conservation of biodiversity, and for this purpose, HNVF indicators play an important role. A fundamental goal of the CAP is the conservation of HNV farmlands and their agricultural systems that are highly multifunctional, contributing to agricultural production while enhancing biodiversity conservation and providing a wide range of ecosystem services [50].

Increasing the socioeconomic viability and appeal of HNV farmlands should be a high priority. To advance HNV farmland management, change needs to be seen as an opportunity rather than as a constraint.

The HNVF index was computed in a GIS environment by applying the statistical approach integrated by the processing of medium resolution remote sensing images. The proposed approach is very versatile because it allows multiple geo-referenced information layers to be managed, and also because it is applicable to different spatio-temporal resolutions (local, regional, and national).

The GIS approach allows the visualization of the individual sub-indices, too, selecting some of them in order to focus on specific problems, and providing high resolution land use, maps of surface temperature, and maps of biomass, as well as chlorophyll, soil, and vegetation moisture indices.

These are some of the most frequently requested pieces of information for the formulation of sustainable management strategies for the landscape–environmental heritage and for the effectiveness of environmental policies [45].

Future works will focus on validations and statistical comparisons of the methodology in Basilicata and in other geographical areas.

**Author Contributions:** Conceptualization, C.F. and F.T.; Methodology, C.F.; Software, A.D.; Validation, C.F. and P.D.; Formal analysis, C.F.; Investigation, P.D.; Data curation, F.T. and F.M.; Supervision, F.M. All authors have read and agreed to the published version of the manuscript.

**Funding:** This research was carried out in the framework of the project 'Smart Basilicata' (2012–2018), which was approved by the Italian Ministry of Education, University and Research (Notice MIUR n.84/Ric 2012, PON 2007–2013 of 2 March 2012), and which was funded with the Cohesion Fund 2007–2013 of the Basilicata Regional authority and 'La casa delle tecnologie emergenti di Matera: il Giardino delle Tecnologie Emergenti', Italian Ministry of Economic Development (MISE).

**Conflicts of Interest:** The authors declare no conflict of interest.

## Appendix A

**Table A1.** Indicators and sub-indicators involved in the HNVF area elaboration by using the statistical approach.

| Index | Sub-Index | Calculation Procedure |
|---|---|---|
| Cultural Diversity (CD) | | *CD = 10 + (1 − C1/UAA * 10)) + (1 − (C2/UAA * 10))* <br> *C1 is the crop area > 10% of the UAA In addition to temporary and permanent forage areas.* <br> $1 \leq CD \leq 10$ |
| Extensive practices (EP) | 2.1. Extensive Managed Crops (EMC) (Weight = 2) <br> 2.2. Soil Moisture Index (SMI) (Weight = 2) <br> 2.3. Extensive Breeding (EB) (Weight = 2) <br> 2.4. Extensive Managed Pastures (EMP) (Weight = 2) <br> 2.5.Nitrogen Surplus (Ns) (Weight = 2) | ***EMC*** *= (Extensive crops + Fallow)(ha)/UAA(ha)* <br> ***SMI*** *derived/rom* <br> ***EB*** *= 1 − Σ(Number of livestock units Surface Temperature and NDVI (MODIS images) * LSU Grazing)/UAA(ha)* <br> ***EMP*** *= Permanent Grassland(ha)/UAA(ha)* <br> ***Ns*** *= Σ(Nfc − Nrc * Rc) * Ac*(i); <br> *c = crop* <br> *Nf = Suggested fertilization* <br> *Nr = nutrient content per unit of biomass of the croop c* <br> *R = Yield of the crop c* <br> *Ac(ha) = area occupied by crop c in cluster i* <br> *(The value of each indicator is between 0 and 1)* |
| Presence of natural elements (Ne) | 3.1 Hedges and stone wall Length (LSM) (Weight = 2) <br> 3.2 Canals and Streams Length (LC) (Weight = 2) <br> 3.3 Lagoons, wetlands, and ponds (SPLS) (Weight = 2) <br> 3.4 Numbers of Lakes (N) (Weight = 2) <br> 3.5 Number of isolated Trees (Nt) (Weight = 2) | ***LSM****= Hedges and dry-stone wall Length(mt)/UAA(ha)* <br> *(if 0 < LSM < 50 mt/ha) LSM = LSM/50* <br> *(if LSM > SO Mt/ha) LSM = l)* <br> ***LC*** *= Canals and Streams Length (mt)/UAA(ha)* <br> *(if 0 < LC < 0.1 mt/ ha) LC = LC/0.1* <br> *(if LC > 0.1 mt/ ha) LC = 1* <br> ***SPLS*** *= Lagoons, wetland and ponds surface(ha)/UAA(ha)* <br> *(if O < SPLS < 0.001 mt/ ha) SPLS = SPLS/0.001* <br> *(if SPLS > 0.001 mt/ ha) SPLS = 1* <br> ***L*** *= Number of lakes/UAA(ha)* <br> *(if 0 < L < 0.003 mt/ ha) L = L/0.003* <br> *(if SPLS > 0.003 mt/ ha) L = 1* <br> ***Nt*** *= Number of isolated trees/UAA(ha)* <br> *(if Nt > 1) Nt = 1* <br> *(The value of each indicator is between O and 1)* |

## Appendix B

**Table A2.** List of the municipalities falling between the 30th and 40th percentile of the HNVF index value. The municipalities, falling between the 30th and 35th percentile, are highlighted in red.

| Municipalities |
|---|
| <span style="color:red">ALIANO</span> |
| BALVANO |
| CASTELLUCCIO INFERIORE |
| <span style="color:red">CERSOSIMO</span> |
| <span style="color:red">EPISCOPIA</span> |
| <span style="color:red">GUARDIA PERTICARA</span> |
| PATERNO |
| <span style="color:red">PIETRAPERTOSA</span> |
| RAPONE |
| <span style="color:red">SALANDRA</span> |
| SAN GIORGIO LUCANO |
| <span style="color:red">TRECCHINA</span> |
| VALSINNI |

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
