# Peer review of "New Technique for Monitoring High Nature Value Farmland (HNVF) in Basilicata"

_sustainability, doi:10.3390/su15108377_

Round 1
Reviewer 1 Report
The manuscript was reviewed, and it found that the authors attempted to identify New Technique for monitoring High Nature Value Farmland (HNVF) in Basilicata. It is a good attempt, and the subject is society's upcoming and present needs.
The work mainly focused on producing HNVF area maps at different spatio-temporal resolutions using a statistical approach and compared with the previous literature work for agreement.
The Abstract explained the concept of nature-valued farmland at large and did not present any significant findings of the work. References (Morelli et al., 2014, De 15 Lucia S., 2013) (European Evaluation Network for Rural Development, 17 2008) were also included in the abstract, which is not a standard format. The study area and methodology adopted are not included in the abstract. The aim and conclusion of the study are incorporated and are in order.
The introduction requires revision. Here first need an introduction to the concept of Nature value farmlands. This information is written in the abstract but needs to be put in the introduction. It will help readers to understand the idea of the work effectively. The rest of the introduction is written well.
The materials and methods are written well with supported literature. The results were presented well with discussion. The cited references were appropriate. I suggest changing the heading of Results to Results and Discussion. The conclusions are written well and are in order.
I appreciate the authors' efforts in developing the NVF index by integrating GIS with a statistical approach.
Author Response
The authors thank the reviewer for appreciating their work and thank him for the suggestions and time spent reviewing the paper.
Please see the attached file which provides a point by point response to the revievwer's comment.
The changes made to the text according to the indications of the reviewers have been highlighted in yellow in the updated version of the paper.

Reviewer 2 Report
its a good paper, few corrections needed
please delete the references from the abstract section
the paper is not according to the guideline of the journal
few grammar in the manuscript please check and correct it
Author Response

(The authors gave the same response as above.)

Reviewer 3 Report
L26 and further: The citations and references (for Sustainability and other MDPI journals) should be numbers in square brackets.
L194: Upper index for “2” in km2 is missing.
L249-L250 and further: Percentile is commonly expressed as the percentage of values in data falling BELOW a given value. Thus, your expression of percentile is possible but counterintuitive and I recommend either explicitly specifying how you present your percentiles (so that the reader is clearly informed) or rewriting it to a standard format.
Figure 4: The caption of the figure should function as separate information for the reader, understandable even without the rest of the text. At least for this image, I doubt that the text can work like this - it's a bit too brief (it would be better to introduce an index a little bit closer, independently of the text).
Figure 4, Figure 5 or “Figure 5”: “Legenda” should be corrected on “Legend”. Also, two different figures are called “Figure 5”.
L336-L337: "Only a qualitative and non-quantitative validation was made due to the large difference between the processing methodologies used to identify HNVF areas." I think the exact opposite is true - because very different methodologies were chosen, it would certainly be useful for the reader to have some spatial correlations calculated between their results - to be able to say which are more similar. Also, this comparison could serve well to rigorously find sites where the different methods lead to very different results. A more detailed study/discussion of those contradictory sites gives the readers the best picture of the validity of the different methods (and it would be possibly a strong argument for using your method).
Table 1: What does the weight mean and why is there a "weight = 2" if it is the same for each variable?
L127-L149: Heretical thought on the end: You have combined as much input data as possible to get the best possible prediction, which is commendable. In statistics, however, a common approach is to fit models that combine simplicity and predictive ability at the same time. If other experts wanted to follow in your footsteps for other regions/countries, it could hypothetically be a problem to have all the data that you had available. I think it would be most useful to try dropping individual variables (or at least sub-indices) from your prediction and comparing the results to the results of "full prediction". The result could be, for example, "if you combine these four specific variables (or you use only this subindex, you will get a result 95% similar to that given by the full model").
English seems to me (although I'm not a native speaker) okay
Author Response
The authors thank the reviewer for appreciating their work and thank him for the suggestions and time spent reviewing the paper.
Please see the attached file which provides a point by point response to the revievwer's comment.

Round 2
Reviewer 3 Report
I am fully satisfied with all the answers of the authors and the changes they made